# Coordinating Effect of VEGFC and Oleic Acid Participates to Tumor Lymphangiogenesis

**DOI:** 10.3390/cancers13122851

**Published:** 2021-06-08

**Authors:** Florent Morfoisse, Fabienne De Toni, Jeremy Nigri, Mohsen Hosseini, Audrey Zamora, Florence Tatin, Françoise Pujol, Jean-Emmanuel Sarry, Dominique Langin, Eric Lacazette, Anne-Catherine Prats, Richard Tomasini, Jean Galitzky, Anne Bouloumié, Barbara Garmy-Susini

**Affiliations:** 1I2MC, Université de Toulouse, Inserm UMR 1297, UPS, 31000 Toulouse, France; florent.morfoisse@inserm.fr (F.M.); fabienne.de-toni@hotmail.fr (F.D.T.); audrey.zamora@inserm.fr (A.Z.); florence.tatin@inserm.fr (F.T.); francoise.pujol@inserm.fr (F.P.); dominique.langin@inserm.fr (D.L.); eric.lacazette@inserm.fr (E.L.); Anne-Catherine.Prats@inserm.fr (A.-C.P.); jean.galitzky@inserm.fr (J.G.); anne.bouloumie@inserm.fr (A.B.); 2CRCM, Inserm UMR 1068, 13001 Marseille, France; jeremy.nigri@inserm.fr (J.N.); richard.tomasini@inserm.fr (R.T.); 3CRCT, Université de Toulouse, Inserm UMR 1037, UPS, 31000 Toulouse, France; hosseini.mohsen7@gmail.com (M.H.); jean-emmanuel.sarry@inserm.fr (J.-E.S.)

**Keywords:** lymphatic, FFA, oleic acid

## Abstract

**Simple Summary:**

In cancer, the lymphatic system is hijacked by tumor cells that escape from primary tumor and metastasize to the sentinel lymph nodes. Tumor lymphangiogenesis is stimulated by the vascular endothelial growth factors-C (VEGFC) after binding to its receptor VEGFR-3. However, how VEGFC cooperates with other molecules to promote lymphatic neovessel growth has not been fully determined. Here, we showed that tumor lymphangiogenesis developed in tumoral lesions and in their surrounding adipose tissue (AT). Interestingly, lymphatic vessel density correlated with an increase in circulating free fatty acids (FFA) in the lymph from tumor-bearing mice. We showed that adipocyte-released FFA are uploaded by lymphatic endothelial cells (LEC) to stimulate their sprouting. Lipidomic analysis identified the monounsaturated oleic acid (OA) as the major circulating FFA in the lymph in a tumoral context. OA transporters FATP-3, -6 and CD36 were only upregulated on LEC in the presence of VEGFC showing a collaborative effect of these molecules. OA released from adipocytes is taken up by LECs to stimulate the fatty acid β-oxidation, leading to increased adipose tissue lymphangiogenesis. Our results provide new insights on the dialogue between tumors and adipocytes via the lymphatic system and identify a key role for adipocyte-derived FFA in the promotion of lymphangiogenesis, revealing novel therapeutic opportunities for inhibitors of lymphangiogenesis in cancer.

**Abstract:**

In cancer, the lymphatic system is hijacked by tumor cells that escape from primary tumor and metastasize to the sentinel lymph nodes. Tumor lymphangiogenesis is stimulated by the vascular endothelial growth factors-C (VEGFC) after binding to its receptor VEGFR-3. However, how VEGFC cooperates with other molecules to promote lymphatics growth has not been fully determined. We showed that lymphangiogenesis developed in tumoral lesions and in surrounding adipose tissue (AT). Interestingly, lymphatic vessel density correlated with an increase in circulating free fatty acids (FFA) in the lymph from tumor-bearing mice. We showed that adipocyte-released FFA are uploaded by lymphatic endothelial cells (LEC) to stimulate their sprouting. Lipidomic analysis identified the monounsaturated oleic acid (OA) as the major circulating FFA in the lymph in a tumoral context. OA transporters FATP-3, -6 and CD36 were only upregulated on LEC in the presence of VEGFC showing a collaborative effect of these molecules. OA stimulates fatty acid β-oxidation in LECs, leading to increased AT lymphangiogenesis. Our results provide new insights on the dialogue between tumors and adipocytes via the lymphatic system and identify a key role for adipocyte-derived FFA in the promotion of lymphangiogenesis, revealing novel therapeutic opportunities for inhibitors of lymphangiogenesis in cancer.

## 1. Introduction

Lymphangiogenesis develops on the edge of solid tumors to provide routes for cancer cells to metastasize to the sentinel lymph nodes and then to distant organs [1]. The activation and proliferation of the lymphatic network in the tumor microenvironment is an orchestrated process that involves cooperation between growth factors and adhesion molecules [2]. However, the tumor stage responsible for the starting signal of the peritumoral lymphangiogenesis remains unclear. The modifications observed in solid tumors during the progression from preneoplastic lesions to adenocarcinoma are sustained by microenvironmental changes, including matrix and cell remodelling [3]. Among these tissues, the adipose depots constitute the major source of energy for the primary tumor and produce various (lymph) angiogenic growth factors, adipokines, hormones, and cytokines that regulate the local microenvironment [4]. Vascular endothelial growth factor-C (VEGFC) was first identified as a multifaceted factor promoting the stimulation of tumor lymphangiogenesis [5]. Then, other growth factors and bioactive peptides were found to participate in that process including VEGFD, VEGFA, FGF2, HGF and apelin [6,7,8]. Most of them are synthesized by adipocytes and are well-known to regulate vessel growth [9]. In addition to protein released from adipocytes, free fatty acids (FFA) generated by adipocyte lipolysis can modulate blood endothelial cell (BEC) function [10,11]. FFAs derive from phospholipase-mediated hydrolysis and are composed by saturated FA (SAFA), monounsaturated FA (MUFA) and polyunsaturated FA (PUFA) [12,13]. FFAs are divided into three types depending upon their amino acid chain lengths: short-, medium-, and long-chain FFAs. Among them, the long-chain saturated palmitic acid (C16:0), the monounsaturated oleic acid (C18:1), and the polyunsaturated linoleic acid (C18:2) comprise the majority of FFA found in serum and in lymph where they are assumed to be the cause of leaky lymphatic vessels in obese subjects [14,15].

FFA are well-known risk factors of cardiovascular diseases [16] and are closely related to the events of metabolic syndromes such as obesity and type 2 diabetes [17,18]. Palmitic acid (PA) impairs endothelial oxidative metabolism, promotes reactive oxygen species (ROS) formation, and decreases cell viability [19]. Despite these detrimental aspects, there are many clinical studies reporting the protective effects of oleic acid (OA) on flow-mediated dilatation and other blood endothelial markers [20]. OA increases lymphatic endothelial cell (LEC) permeability [21] and was proposed as a solute for lymphatic delivery of nanoparticles [22] and poor water-soluble drugs [23]. In cancer patients, high FFA level is observed in the serum and may represent an indicator of cancer suggesting their role in tumor progression [24]. In particular, OA strongly decreases oxidative stress in melanoma [25]. FFAs are transported into cells by fatty acid transporter proteins composed of six isoforms (FATP1-6) and the fatty acid translocase/cluster of differentiation 36 (FAT/CD36). After being uploaded by cells, FFAs are converted into metabolites, which participate in a variety of cellular regulatory mechanisms as second messengers. Recent findings highlighted that the beta-oxidation of intracellular fatty acids (FA) is crucial for LEC metabolism, but nothing is known about the role of lymph-circulating free FA on LEC function [26].

In this study, we showed that lymphangiogenesis develops in preneoplastic lesions and in the peritumoral adipose tissue. This lymphangiogenesis is dependent on VEGFR-3 signaling and is associated with an increase in circulating FFA in the lymph. We showed that adipocyte-released FFA can be loaded by LECs to stimulate their sprouting, independently of growth factors. We identified the monounsaturated OA as a key regulator of the lymphangiogenesis in cancer. OA in association with VEGFC stimulated the expression of the transporters FATP-3 and -6 and CD36 at the surface of the lymphatic endothelial cells to activate their metabolism and the production of reactive oxygen species (ROS). In the past decade, studies have hinted at important connections between adipose depots and lymphatic vessel function, but clear molecular links have not been established. Here, we reported that tumor lymphangiogenesis is not only promoted by the canonical growth factor/tyrosine kinase receptor pathway, but involves a cooperation with lipids, in particular adipocyte-released FFA to improve the lymphatic vessel growth. We demonstrated that VEGFC and OA work together to stimulate the peritumoral lymphatic growth thus providing a reliable network for metastases. Altogether, this work provides new critical insight for the development of anti-lipolysis therapies to reduce tumor lymphangiogenesis and lower the risk of metastasis.

## 2. Results

### 2.1. Lymphangiogenesis Develops in Preneoplastic Lesions and in Peritumoral Adipose Tissue

To study the lymphangiogenesis in the tumor microenvironment, we first investigated the lymphatic system in human pancreatic ductal adenocarcinoma (PDAC) characterized by a strong desmoplastic reaction that prevents the development of blood vessels [27]. However, we identified an extensive infiltration of lymphatic vessels into the stroma, which is in accordance with the fact that lymphatics develop in collagen-rich tissues [28]. The lymphangiogenesis started in preneoplastic lesions (PanIN) (Figure 1A) extending into the vicinity of peritumoral adipose tissue (AT) (Figure 1B) and in draining lymph nodes (Appendix A). To confirm the expansion of the lymphatic network in peritumoral AT, we used three mouse models of cancer, including mice with transgenic pancreatic adenocarcinoma (Pdx1Cre; LSLKras^G12D^Ink4a^+/−^; PKI mice) [29], melanoma B16 (B16) and Lewis Lung Carcinoma (LLC) (Figure 1C–H) [30]. As observed in human tissues, we found a stimulation of lymphangiogenesis in the PKI mice starting in preneoplastic lesions (PanIN3) and developing in peritumoral AT (Figure 1C,D). Interestingly, we also found lymphangiogenesis in both tumors (Figure 1E,F) and peritumoral AT (Figure 1G,H) from the melanoma B16 and LLC tumor-bearing mice. This was associated with increased systemic inflammation in LLC and B16 tumor models as previously described (Appendix A) [31].

### 2.2. Blocking Preneoplastic Lesion Lymphangiogenesis Improves Survival

To study the molecular mechanisms that control the peritumoral lymphangiogenesis, we used a blocking antibody against VEGFR-3 (mF4-31C1) and VEGFR-2 (DC101) to inhibit lymphangiogenesis and angiogenesis, respectively, (Figure 2) [32]. We generated Kaplan–Meier survival curves for the antibody-treated PKI mice (Figure 2A). Without treatment these mice have a median survival of nine weeks [29]. We found that the anti-VEGFR-3 treatment improved the survival of PKI mice significantly, when compared to anti-VEGFR-2 or an isotype control IgG-treated mouse (Figure 2A). As PKI mice rapidly develop ADK, we next used heterozygous mice that have a median survival of four months [29] and thus allowed the treatment at the preneoplastic stages (17 to 20 weeks). We observed a significant improvement compared to PKI mice using the blocking VEGFR-3 as survival was increased threefold, thus showing that blocking lymphatic vessel growth in preneoplastic lesions may be pivotal to improve survival rates (Figure 2B). We next treated B16- and LLC bearing-mice with the VEGFR-3 blocking antibody. We observed an inhibition of tumor lymphangiogenesis (Figure 2C,D) that was associated with a decrease in tumor metastasis into the sentinel lymph nodes (Figure 2D–F), confirming that blocking peritumoral lymphangiogenesis inhibits metastasis.

### 2.3. Blocking Peritumoral AT Lymphangiogenesis Decreases Lymph Circulating Lipid Levels

Next, peritumoral AT from LLC and B16 tumor-bearing mice treated with the anti-VEGFR-3 blocking antibody were collected and analyzed. We observed an inhibition of peritumoral AT lymphatic density showing that the inhibition of lymphangiogenesis was not restricted to tumors (Figure 3A–C).

We next performed an intra-lymphatic injection of oil-diluted lectin-FITC to analyze the connection between the lymphatics and tumor (Appendix A). The blocking antibody abolished lymphatic transport to the tumor, thus showing that transport from AT to the tumor was blocked (Appendix A). In parallel, to investigate the role of peritumoral AT lymphatic vessels, dye fluorescent lipid (Bodipy) was injected into the lymphatic system (Figure 3D,E). The fluorescent lipid leaked out of the lymphatics into the peritumoral AT showing the local plasticity and permeability of lymphatics. Interestingly, fluorescent lipids spread to distant adipose depots demonstrating an increased lipid circulation in the lymph from tumor bearing mice compared to wild- type mice (Figure 3E,F). This was reduced in the presence of the blocking antibody, revealing an active control of lymphatic vessels in the peripheral lipid transport during tumor progression (Figure 3E,F). Interestingly, we also observed an increase of circulating lipase in the lymph that was reduced by the anti-VEGFR-3 blocking antibody (Appendix A). This effect was related to inhibition of fat depot loss, as determined by measuring the total fat content by echo-MRI (Appendix A) or weighing of the LLC distant AT (Appendix A). Here, we showed that lipids are conveyed from tumor proximal to distant adipose depots through the lymphatic system by an active mechanism that can be inhibited by the VEGFR-3 blocking antibody.

### 2.4. Tumor-Bearing Mice Exhibit Increased Level of Circulating Free Fatty Acids in Lymph

To identify protein biomarkers that contribute to the peritumoral lymphangiogenesis, we performed a comprehensive proteomic analysis of the lymph and serum proteomes of the PKI mice (Figure 4). Lymph and serum exhibited different migration profiles on acrylamide gels (Figure 4A). No significant difference in total protein amount was shown between control and tumor-bearing mice. However, we identified three to five times more proteins in the lymph compared to serum, mostly due to the identification of degraded peptides from extracellular space that are drained by lymphatics (Figure 4B). The majority of proteins identified in lymph compared to serum were involved in lipid metabolism, suggesting modifications in surrounding adipose tissue (Figure 4C). Notably, we observed an upregulation in lymph, but not in serum, of apolipoproteins (Apo), which perform lipid transport (Appendix A). We identified three times more changes in protein amounts involved in FFA metabolic processes in lymph (Figure 4D) compared to serum (Figure 4E) in tumor-bearing mice.

### 2.5. Oleic Acid Released by Adipose Tissue Drive LEC Sprouting

We next identified which FFA transport was modified in tumor-bearing mice lymph (Figure 5). Adipose tissue (AT) is a key organ in the regulation of total body lipid homeostasis, which is responsible for the storage and release of FFA according to metabolic needs including saturated FA (SAFA), monounsaturated FA (MUFA) and polyunsaturated FA (PUFA) [12,13]. To determine which FFAs were conveyed by lymphatic vessels, lipidomic analysis of the lymph from B16- and LLC tumor-bearing mice treated with the VEGFR-3 blocking antibody was performed (Figure 5A–C). We found that lymph levels of long chain saturated, mono- and poly-unsaturated lipids, which are major substrates for mitochondrial β-oxidation, were increased in lymph from tumor-bearing mice and reduced in the mice after treatment with VEGFR-3 blocking antibodies (Figure 5B,C) [33]. The most abundant FFAs in the lymph from tumor-bearing mice were oleic acid (OA) and palmitic acid (PA). Their levels were decreased by the anti-VEGFR-3 treatment, suggesting that the decrease in lymphatic vessel density in the adipose tissue reduces the FFA transport (Figure 5C).

We next examined which FFA transporters are expressed on LECs. We confirmed that LECs express FA transport proteins (FATPs) and FA translocase (CD36) (Figure 5D,E, Appendix A) [34]. In particular, we identified a significant upregulation of FATP3 and FATP6 after OA+VEGFC stimulation, suggesting a cooperative effect of these molecules on FFA transporter expression (Figure 5D). Interestingly, only VEGFC was able to stimulate CD36 expression in LECs (Figure 5E). Altogether, these data suggest that VEGFR-3 signaling is necessary for the increased cellular FA uptake by LECs.

### 2.6. Lymphatic FFAs Stimulate LEC Function

To determine whether AT-released FFAs could control lymphangiogenesis in association with tumor cell-released growth factors, we next performed proliferation and tube forming assays on LECs incubated with conditioned media (Figure 6). Pancreatic tumor cells (Figure 6A,B), as well as B16 and LLC (Figure 6C,D), promoted both LEC proliferation and sprouting, whereas adipocyte conditioned media (AM) only promoted branch point formation suggesting distinct signaling pathway stimulation (Figure 6E,F). To avoid any effect due to lymphangiogenic growth factors released from adipocytes, heat-inactivated adipocyte conditioned media (heated-AM) were produced. Heated-AM stimulated LEC branch point formation, but not proliferation in the same extent as AM suggesting that non-peptidic molecules released from adipocytes can stimulate the LEC function (Figure 6G,H).

The main function of adipocyte is to store lipids and triglycerides that are hydrolyzed by lipase to generate the release of free fatty acids (FFA) and glycerol. To investigate the role of lipolysis-generated FFA on LEC, adipocyte lipolysis was induced by beta-adrenergic receptor stimulation using isoproterenol (ISO) and forskolin (FK) (Figure 6I–L). FFA and glycerol release from adipocytes was measured as a control of lipolysis (Appendix A). To determine whether FFA could activate LECs, we incubated cells with BIODIPY, a neutral lipid that serves as tracer for other lipids. In presence of heated-AM, BIODIPY accumulated into LECs more than in VEGFC- or AM-stimulated LECs showing the uptake of FFA by LECs (Figure 6J,K). The FFA uptake was increased in the presence of lipolytic media (Figure 6I,J) as well as LEC sprouting (Figure 6K). The effect of FFA on LEC was confirmed as LEC sprouting was reduced in the presence of conditioned media from lipolytic adipocytes treated with hormone sensitive lipase (BAY), and non-selective lipase (E600) inhibitors showing the selective and modular effect of FFA on LEC (Figure 6L, Appendix A).

### 2.7. Exogenous Oleic Acid Stimulates LEC Beta-Oxidation

To confirm that FFA uptake stimulates lymphangiogenesis, we next performed lymphatic sprouting assay. The uptake of OA by LECs stimulated LEC sprouting (Figure 7A). In contrast, PA induced LEC cell death [35], therefore suggesting that the beneficial effect of FFA on the lymphatic endothelium is restricted to OA (FA chain length C18). These results were validated by FATP3 and FATP6 siRNA transfection which significantly decreased OA-induced branch point formation in LECs, confirming the crucial role of these transporters in LEC function during lymphangiogenesis induced by VEGFC + OA (Figure 7A). Lymphatic development is dependent on intra-cellular fatty acids beta-oxidation [26]. To investigate whether extra-cellular FFA could also participate in that metabolic process after being uptaken by LECs, we next performed in vitro blockade of mitochondrial FA entry with the CPT1a inhibitor, etomoxir (Eto). Eto inhibited OA + VEGFC-induced LEC branch formation (Figure 7B), thus confirming that exogenous FFAs could also stimulate the LEC metabolic function. To analyze if the bioenergetic status of LECs is involved in FFA-induced cell functions, we characterized mitochondrial ATP production and oxygen consumption rates (OCR) of OA and/or VEGFC-stimulated LECs (Figure 7C–E). We found that ATP-linked OCR is increased by OA and this is reduced in the presence of etomoxir or anti-VEGFR-3 (Figure 7C–E). Interestingly, quantification of mitochondrial reactive oxygen species (mtROS) revealed increased levels of mtROS in VEGFC + OA-stimulated LECs, which was inhibited by the VEGFR-3 blocking antibody (Figure 7F) and the N-acetyl cysteine (NAC) scavenger of ROS (Figure 7G), showing that the stimulation of LEC function by OA+VEGFC is dependent on the activation of LEC mitochondrial activity. Importantly, the effect of exogenous OA is dependent on VEGFR-3 activation by VEGFC to be uploaded and to stimulate LEC sprouting.

## 3. Discussion

The development of solid tumors is often preceded, both in humans and experimental animal models, by the appearance of lesions referred to as preneoplastic. Here, we observed the development of lymphatic vessels in preneoplastic lesions suggesting that tumors sustain an early favorable environment for metastasis to distant loci. Additionally, we showed that AT supports this lymphangiogenesis that is consequently independent of the molecules produced by the adenocarcinoma itself. We confirmed the importance of this early lymphangiogenesis as blocking VEGFR-3 pathways in preneoplastic lesions and surrounding AT showing improved survival and decreased metastases. Adipocytes produce both angiogenic and lymphangiogenic molecules. In humans, tumor masses are significantly larger in obese patients and the tumor vessel density inversely correlates with the adipose area, leading to a resistance of obese patients to anti-VEGF therapy [36]. Multiple factors modulate the complex interplay between the vascular system and adipocytes, therefore targeting the adipose lymphatic vasculature may provide new therapeutic options not only for treatment of obesity and metabolic disorders, but for treatment of cancer [37]. VEGFC and -D are elevated in adipose tissue during obesity [38] and an overexpression of VEGFD in adipose tissue resulted in de novo lymphatics and improved overall metabolism in mice [38]. Adipocytes also produce peptidic molecules that are able to stimulate the lymphatic endothelial function such as apelin [7]. Secreted molecules are then collected by the lymphatic system, making the lymph composition representative of the environmental metabolic status. The lymph composition is dependent on the ultrafiltration of plasma proteins as well as proteins and molecules derived from the metabolic and catabolic activities of surrounding tissues [39]. Therefore, the lymph protein and lipidic composition conveys major information on environmental tissue metabolic activity. Here we showed that unsaturated long-chain FA are prominent compounds of the lymph in cancer and their transport is in part regulated by the lymphatic activity. Among them, OA and PA represent the largest FFA identified. Most FFA’s have deleterious effect on the endothelium function [40]. Their levels are increased in subjects with obesity and type 2 diabetes, playing detrimental roles in the pathogenesis of atherosclerosis and cardiovascular diseases [40]. In blood endothelium, FATP3 and FATP4 are required for FFA transport across the vascular endothelial barrier [41]. Here, we observed a difference in blood and lymphatic endothelium as LEC expressed both FATP3 and 6 initially described to mediate fatty acid uptake in cardiomyocytes [42]. The mechanism of how FFAs modulate LECs remains unclear. LEC metabolism involves intracellular FA beta-oxidation. Here we showed that circulating FFA also promotes this process, suggesting a new way for adipose tissue to control lymphatic endothelial physiology. Conditioned media from lipase inhibitor-treated adipocytes significantly reduced LEC sprouting and migration, demonstrating the crucial role of FFAs in the stimulation of lymphatic function. Here, we confirmed the deleterious effect of PA (palm oil) on the endothelium, whereas we observed an opposite effect of OA (olive oil) on lymphatic compared to blood endothelium [40]. Whereas OA is known to diminish ATP-induced mobilization in bovine aortic endothelial cells (BAECs) [43], we observed an increase in ATP production and oxygen consumption rates in LEC showing the importance to selectively study the lymphatic transport of FFA in tumoral conditions. Lymphatic vessels are rare in the normal AT but often more conspicuous in tumors surrounding AT. However, the understanding of the mechanisms controlling fat transport and drainage of peripheral AT by the lymphatic system has remained limited to the study of intestinal villi. Our findings provide evidence for a new level of crosstalk between AT and lymphatic drainage to promote tumor lymphangiogenesis. Our work also provides novel therapeutic perspectives for the use of VEGFR-3 blocking antibodies in the regulation of circulating FFA levels. It suggests that targeting the lymphatic system could represent a promising strategy for treating metabolic diseases in which FFA release is stimulated, such as obesity-associated type 2 diabetes.

## 4. Material and Methods

### 4.1. Human Tissues

In total, 15 primary human pancreatic adenocarcinoma specimens and their associated lymph nodes were collected. Samples were obtained from archival paraffin blocks of pancreatic cancer from patients treated at the Rangueil hospital, Toulouse, France between 2002 and 2008. Samples were selected as coded specimens under a protocol approved by the INSERM Institutional Review Board (DC-2008-463) and Research State Department (Ministère de la recherche, ARS, CPP2, authorization AC-2008-820) and included tumor specimens identified as pancreatic adenocarcinoma. All tumor and sentinel lymph node specimens were obtained from pre-treatment surgical resections. Final pancreatic cancer diagnosis was confirmed by pathologic evaluation of specimens stained with hematoxylin and eosin (H&E). Each series included as controls normal pancreatic tissue and selected common specimens to ensure consistent interpretation.

### 4.2. Animal Study

All studies received local ethics review board approval and were performed in accordance with the guidelines of the European Convention for the Protection of Vertebrate Animals used for experimentation and according to the INSERM IACUC (France) guidelines for laboratory animal husbandry (CEEA-122 2015-16). All animal experiments were approved by the local branch Inserm Rangueil-Purpan of the Midi-Pyrénées ethics committee, France. Animals from different cages in the same experimental group were selected to assure randomization. Mice were housed in individually ventilated cages in a temperature and light regulated room in a SPF facility and received food and water ad libitum. Female C57BL/6J (6 weeks old) were obtained from Janvier (Le Genest Saint Isle, France). Pdx1-Cre;LSL-Kras^G12D/+^; Ink4a^−/+^(PKI) and Pdx1-Cre;LSL-Kras^G12D/+^; Ink4a^−/−^ mice were generated as described previously [29]. Briefly, the mutation KrasG12D or the inactivation of Ink4a/Arf failed to produce any neoplastic lesions in the pancreas. In combination, KrasG12D expression and Ink4a/Arf deficiency resulted in an earlier appearance of PanIN lesions and these neoplasms progressed rapidly to highly invasive and metastatic cancers similar to the human disease with a proliferative stromal component and ductal lesions.

Tumor models. 5 × 10^5^ LLC or B16 melanoma cells or saline solution as negative control were injected subcutaneously (close to inguinal fat pad) into female C57BL/6J mice to generate ectopic or orthotopic xenograft models, respectively. After 18 and 21 days, mice were anaesthetized with a ketamin–xylasine solution injected intraperitoneally (i.p). The mice were then sacrificed. Tissues and tumor were excised, weighed, embedded into OCT compound (Tissue-Tek; Sakura Finetek, Torrance, CA, USA) and frozen at −80 °C.

### 4.3. Cell Culture

Mouse B16K1 (MHC class I–positive B16F10) (B16) melanoma cell line was used as previously described [44]. Mouse Lewis lung carcinoma (LLC) cell line was purchased and authenticated from American Type Culture Collection. These murine cell lines were cultured in Dulbecco’s Modified Eagle Medium (DMEM) high glucose cell culture medium containing 10% fetal bovine serum (FBS) and were maintained at 37 °C in a humidified atmosphere. Human dermal lymphatic endothelial cells (HDLECs) were purchased from Promocell (Promocell, Heidelberg, Germany), cultured in ECGM-MV2 (Promocell, Heidelberg, Germany) medium and used between passages 2 and 6.

### 4.4. Chemical and Reagents

Human recombinant VEGFC was from R&D systems (Minneapolis, MN, USA). Oleic acid (OA), palmitic acid (PA), etomoxir, Nile Red was purchased from Sigma–Aldrich (Saint-Quentin Fallavier, France). Rabbit anti-mouse LYVE-1 antibody was from Fitzgerald (Acton, MA, USA), mouse anti-human podoplanin (D2/40) from DAKO (DAKO, Carpinteria, CA, USA), and goat anti-PAN cytokeratin from Santa Cruz Biotechnologies (Dallas, TX, USA). Etomoxir is an inhibitor of the carnitine palmitoyl transferase I (CPT1a) that initiates the mitochondrial oxidation of long-chain fatty acids. It is the key enzyme in the carnitine-dependent transport across the mitochondrial membrane leading to fatty acid beta-oxidation. Etomoxir is an irreversible inhibitor of the CPT1 enzyme that decreases β oxidation in the mitochondria.

### 4.5. Immunohistochemistry

Tumors, lymph nodes and adipose tissues were embedded into OCT compound and 5 µm tissue sections were immunostained with specific antibodies. The mean number of lymphatic vessels was quantified in 5–10 microscopic fields per cryosection using automated pixel density determination. The mean number of mice with metastases in draining lymph nodes was determined by immunostaining cryosections with 10 μg/mL of anti-pancytokeratin (Santa Cruz Biotechnologies, Dallas, TX, USA). Immunostaining was quantified in 5–10 microscopic fields per cryosection using automated pixel density determination as the mean number of pixels ± SEM for each group.

### 4.6. Labeling of the Lymphatic Compartment in AT

To assess the lymphatic vasculature in peritumoral AT, LLC- and B16-bearing mice treated with control IgG or anti-VEGFR-3 were anaesthetized with a ketamin–xylasine solution injected i.p and then 20 µL of DyLight 488 Lycopersicon Esculentum Lectin (Vector Laboratories) were injected into the footpad (hind foot, side of the tumor) 10 min prior to euthanasia. Peritumoral adipose tissue was excised, weighed and stained with Red Oil (Sigma–Aldrich, Saint-Quentin Fallavier, France).

### 4.7. In Vivo Treatment with Anti-VEGFR2 and VEGFR3 Blocking Antibodies

LLC- and B16-bearing mice were injected intraperitoneally with 200 µg of blocking rat anti-mouse VEGFR-2 (DC101), or anti-VEGFR-3 (mF431C1, all from Lilly Pharmaceutical), or control IgG (Affymetrix-eBioscience). To determine the impact of VEGFR-2 and R-3 blockade on PKI heterozygous mice survival, Pdx1-Cre; LSL-Kras^G12D/+^, Ink4a^−/+^ were injected with 200µg of control IgG or anti-VEGFR-2 (DC101) or anti-VEGFR-3 (mF431C1) antibody every three days from week 17 to week 20 (*n* = 6 animals per group). To determine the impact of VEGFR-2 and R-3 blockade on PKI homozygous mice survival, Pdx1-Cre; LSL-Kras^G12D/+^; Ink4a^−/+^ were injected with 200 µg of control IgG or anti-VEGFR-2 or anti-VEGFR-3 antibody every three days from week 6 to week 8 (*n* = 6 animals per group).

### 4.8. Lymph Collection

Lymph nodes were excised and the capsule was carefully dissected in 50 μL PBS. After gentle centrifugation so as not to destroy immune and stromal cells, supernatant (lymph) was retrieved and frozen at −80 °C.

### 4.9. ProteoMiner Protein Equalization and Analysis of Peptides by LC-MS/MS

For proteomic analysis, we used shotgun proteomics that combine technologies to identify peptides produced by proteolytic digestion of proteins [45] coupled to ProteoMiner (Bio-Rad Laboratories Ltd., Hercules, California, USA). that provide random generation of hexapeptides and creates a substantial ligand library for which proteins can selectively bind [46]. Briefly, 700 μL lymph and serum (*n* = 9 mice) protein homogenate were concentrated before application to the ProteoMiner low-yield enrichment kit followed by elution and loading 1-D SDS-PAGE gel (Bio-Rad Laboratories Ltd.). The resulting peptides were extracted from the gel and analyzed by nano-LC-MS/MS using an Ultimate 3000 system (Dionex, Sunnyvale, California, USA) coupled to an LTQ-Orbitrap Velos mass spectrometer or LTQ- Orbitrap XL (Thermo Fisher Scientific, France). Bioinformatic analysis was performed using ingenuity pathway analysis software. Results were normalized per mg of total proteins.

### 4.10. Free Fatty Acid Methyl Ester (FAME) Analysis

Lipids corresponding to 10–50 µg of lymph’s proteins (from control, B16 and LLC mice) were extracted according to Bligh and Dyer [47] in dichloromethane/methanol/water (2.5:2.5:2.1, *v/v/v*), in the presence of the internal standards heptadecanoate acid (2 µg). The lipid extract was directly methylated in boron trifluoride methanol solution 14% (SIGMA, 1 mL) and heptane (1 mL) at RT for 10 min. After the addition of water (1 mL) to the crude, FAMEs were extracted with heptane (3 mL), evaporated to dryness and dissolved in ethyl acetate. FAME were analyzed by gas–liquid chromatography [48] on a Clarus 600 Perkin Elmer system using a Famewax RESTEK fused silica capillary columns (30 m × 0.32 mm i.d, 0.25 μm film thickness). Oven temperature was programmed from 110 °C to 220 °C at a rate of 2 °C per min and the carrier gas was hydrogen (0.5 bar). The injector and the detector were at 225 °C and 245 °C, respectively.

### 4.11. Sprouting Assay

To perform tube formation assay, twenty-four well plates were coated with 300 µL of growth factor reduced matrigel (Corning, 354230) and incubated for 30 min in 37 °C to allow gelling. 5 × 10^4^ HDLEC were seeded per well in the appropriated medium at a ratio 1:1 with EBM2 containing 0.5% of FBS with or without etomoxir (50–300 µM). At least 5 pictures per well were taken after 8 and 24 h hours with a phase-contrast optical microscope using a 5× objective (Leica, DMi8). The number of branch points was quantified using ImageJ software.

### 4.12. Heated Adipocyte Medium

To eliminate growth factors from adipocyte media, media were heated for 10 min at 95 °C.

### 4.13. Bodipy and Red Oil Uptake Assay

HDLECs were seeded on coverslip and cultured for 12 h in EGM2-MV medium. To explore the effect of adipocytes conditioned medium on lipid uptake, cells were incubated with the tested medium at a 1:1 ratio with EBM2 (Lonza, Basel, Suisse) overnight with or without etomoxir (100 µM) or FK+Bay and FK+E600 conditioned media. Cells were then washed twice with PBS and incubated for 10 min in 37 °C with PBS-BSA1%, 20 µM C1-Bodipy-C12 (Sigma Aldrich, Alsace, France) or red oil (Sigma Aldrich). HDLECs were washed vigorously with PBS and fixed for 10 min in PFA 4% at room temperature. After 5 min in 1:10,000 DAPI reagent to label nuclei, coverslips were washed with PBS and mounted with fluorescent mounting medium (DAKO, Carpinteria, CA, USA) under slides. For each experiment at least 6 frames were photographed using a 40× objective (Leica, Wetzlar, Germany, DMi8).

### 4.14. Quantitative Real-Time RT-PCR

Total cellular RNA was isolated from human HDLECs using RNAqueous^®^-Micro Kit (Ambion, TX, USA) according to the manufacturer’s instructions. A total of 100 ng RNA was used to synthesize cDNA using SuperScript^®^ VILO cDNA Synthesis Kit (Ambion, TX, USA). The expression of human FAT/CD36 (fatty acid translocase) and fatty acid transport proteins (FATP1, FATP2, FATP3, FATP4 and FATP6) was investigated by SYBR green real-time reverse transcribed polymerase chain reaction using the ABI StepOne+ real-time PCR system (Applied Biosystems, Villebon s/Yvette, France). Each reaction was run with 18S as a reference gene and all data were normalized based on the expression levels of 18S. The following primer sequences were used: Fatp/CD36, 5′-AAATAAACCTCCTTGGCCT-3′ forward, 5′-GCAACAAACATCACCACACC-3′ reverse; Fatp3, 5′- CAGAGACCTTCAAACAGCA-3′ forward, 5′-CAGAACGTACAGTGGTCA-3′ reverse; Fatp6, 5′-GCGTGGTGGCCTTTCTCA-3′ forward, 5′-ACAGGCGCGGATGCAAT-3′ reverse; FATP3 siRNA is a set of 4 SMARTpool ON-TARGETplus siRNA from Dharmacon (LQ-007499-01-0002). Targeted sequences: CCUUGAUUCGCUAUGAUGU; CUGCGAUGACCAAGGUUUU; GGAUCAGGGAUGUUUGCGA; UCCGCUUCCAUGAUCGUAC; FATP6 siRNA is a set of 4 SMARTpool ON-TARGETplus siRNA from Dharmacon (LQ-007502-01-0002). Targeted sequences: CCUUAACUGGAGACUUA; GGUAUAUGAUUCUUUAUGA; GAGCUAACAUUAAUUAUGC; CCACUUUACUUCAUGGAUA.

### 4.15. MitoSOX Red

Mitochondrial Superoxide Indicator (Thermofisher, Illkirch-Graffenstaden, France) was incubated according to manufacturer instructions. Briefly, LECs were incubated 15 min at 37 °C in the presence of Oleic acid and Palmitic acid stimulation (300 μM).

### 4.16. Seahorse Assays

Metabolic analyses in HDLECs were performed with a Seahorse XFe96 analyzer (Agilent Seahorse, CA, USA) according to the manufacturer’s recommendations. In brief, OA (300 μM) + VEGFC (50 ng/mL) stimulated LECs (150,000 cells per well of a 24-well plate) were maintained in a CO_2_ incubator for 2 h. Then, LECs were incubated with etomoxir or anti-VEGFR3 for 2h A Glycolysis Stress Test kit (Agilent Seahorse, CA, USA) was used to monitor the extracellular acidification rate under various conditions. Three baseline recordings were made, followed by sequential injection of glucose (10 mM), the mitochondrial/ATP synthase inhibitor oligomycin (3 μM), and the glycolysis inhibitor 2-deoxy-d-glucose (2-DG; 100 mM).

### 4.17. Statistics

Statistical analysis was performed using Prism 5 with 2 ways unpaired T test to detect significant differences between 2 experimental groups or 2-way ANOVA to compare multiple conditions in cell analyses.

## 5. Conclusions

Our findings provide evidence for a new level of crosstalk between AT and lymphatic drainage to promote tumor lymphangiogenesis. Our work also provides novel therapeutic perspectives for the use of VEGFR-3 blocking antibodies in the regulation of circulating FFA levels. It suggests that targeting the lymphatic system could represent a promising strategy for treating metabolic diseases in which FFA release is stimulated, such as obesity-associated type 2 diabetes.

## Figures and Tables

**Figure 1 cancers-13-02851-f001:**
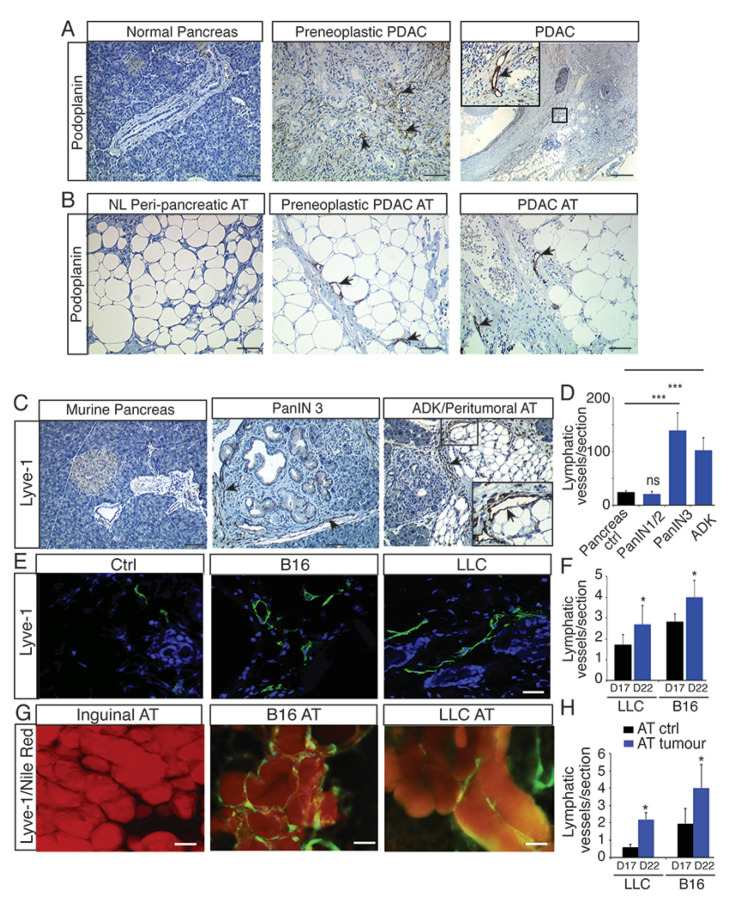
Lymphangiogenesis develops in primary tumors and its surrounding adipose tissue. (**A**) Representative images of podoplanin labelling in a normal human pancreas and in pancreatic ductal adenocarcinoma (PDAC) at preneoplastic and carcinoma stages (scale bar, 100 μm) (*n* = 15). (**B**) Representative images of podoplanin labelling in normal and peritumoral adipose tissue (AT) from human preneoplastic and PDAC (scale bar, 100 μm) (*n* = 15). (**C**) Representative images of Lyve-1 labelling in normal murine pancreas, PanIN3 and adenocarcinoma stages in Pdx1Cre; LSLKrasG12DInk4a^+/−^ (PKI) mice (scale bar, 50 μm). (**D**) Quantification of the lymphatic vessel density in tumors from PKI mice at preneoplastic (PanIN) and carcinoma stages (mean + sem; *n* = ten animals; *** *p* < 0.005, ns: not significant). (**E**) Representative images of Lyve-1 labelling in melanoma B16 (B16) and Lewis Lung Carcinoma (LLC) tumor models (scale bar, 50 μm). (**F**) Quantification of lymphatic vessel density in LLC and B16 tumor models (* *p* < 0.001, *n* = ten animals). (**G**) Representative images of Lyve-1 and Nile red labelling in LLC and melanoma B16 peritumoral adipose tissue (scale bar, 50 μm). (**H**) Quantification of the lymphatic vessel density in LLC and B16 peritumoral adipose tissue (mean + sem; *n* = ten animals; * *p* < 0.005).

**Figure 2 cancers-13-02851-f002:**
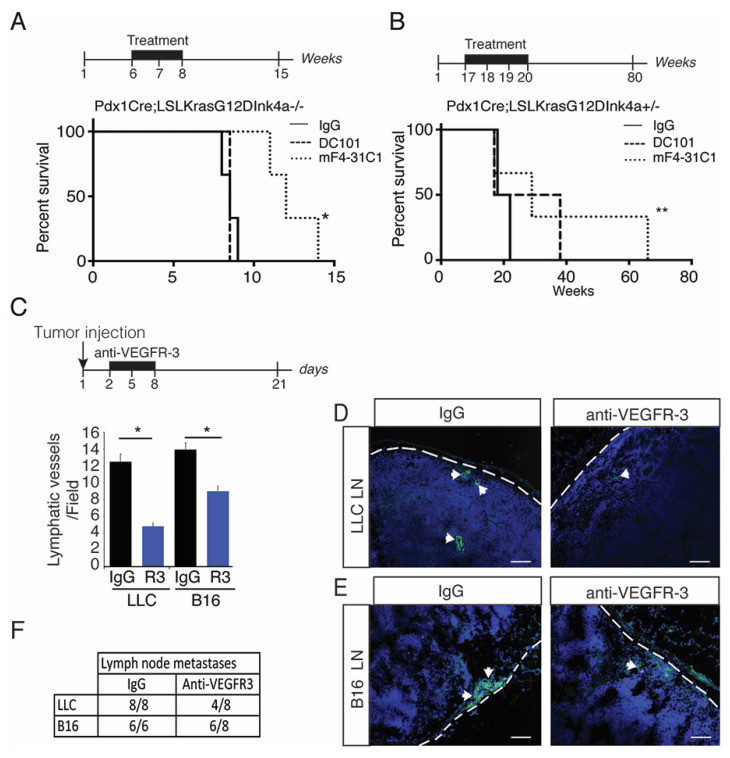
Blocking tumor lymphangiogenesis increases survival and reduces metastasis. (**A**) Kaplan–Meier survival curves of Pdx1Cre; LSLKrasG12DInk4a^−/−^ mice treated with anti-VEGFR-2 (mF4-3C1) or anti-VEGFR-3 (DC101) antibodies; * *p* < 0.05. (**B**), Kaplan–Meier survival curves of PKI mice treated with anti-VEGFR-2 or anti-VEGFR-3 antibodies; ** *p* < 0.01. (**C**) Quantification of the lymphatic vessel density in LLC and B16 tumors treated with IgG control isotype or the anti-VEGFR-3 (mean + sem; *n* = ten animals; * *p* < 0.05). (**D**,**E**) Representative images of cytokeratin labelling in LLC (**D**) and B16 (**E**) draining lymph nodes (scale bar, 50 mm). (**F**) Quantification of lymph node metastases in LLC- and B16-tumor bearing mice.

**Figure 3 cancers-13-02851-f003:**
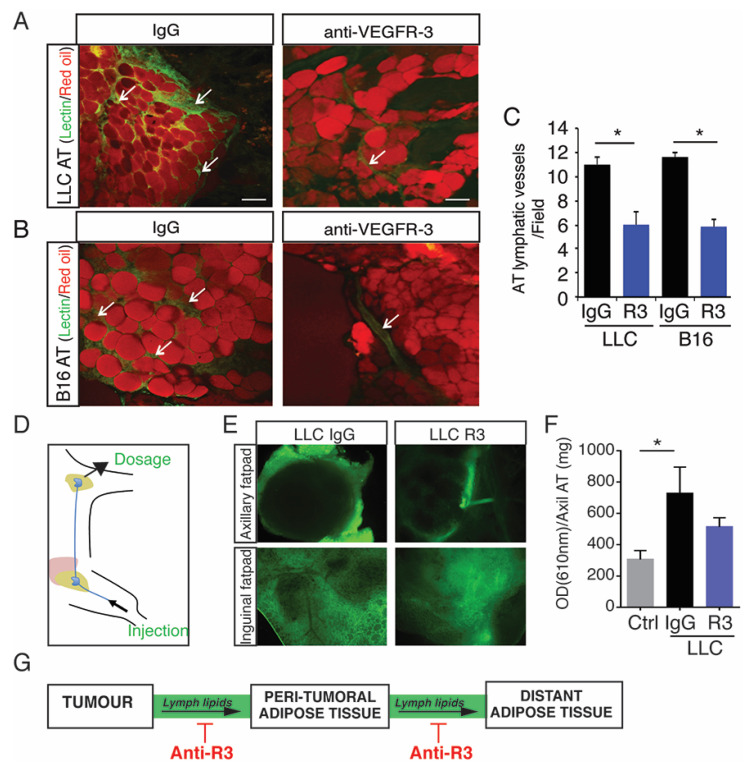
Blocking AT lymphangiogenesis inhibits lymph systemic lipid transport. (**A**,**B**) Representative images of FITC-lectin (arrows) and red oil labelling in AT from LLC (**A**) and B16 (**B**) tumor-bearing mice treated with IgG control isotype or the anti-VEGFR-3 (scale bar, 50 μm). (**C**) Quantification of the lymphatic vessel density in peritumoral AT from LLC and B16 tumor-bearing mice treated with IgG control isotype or the anti-VEGFR-3 (mean + sem; *n* = ten animals; * *p* < 0.05). (**D**) Schematic representation of lipid-diluted fluorescent dye in the lymphatic system from tumor-bearing mice. (**E**) Inguinal and axillary adipose depots from LLC tumor-bearing mice treated with IgG control isotype or the anti-VEGFR-3 (scale bar, 50 μm). (**F**) Quantification of fluorescence intensity in axillary fat depot (* *p* < 0.05). (**G**) Schematic representation of the anti-VEGFR3 antibody action on peripheral lymphatic lipid transport.

**Figure 4 cancers-13-02851-f004:**
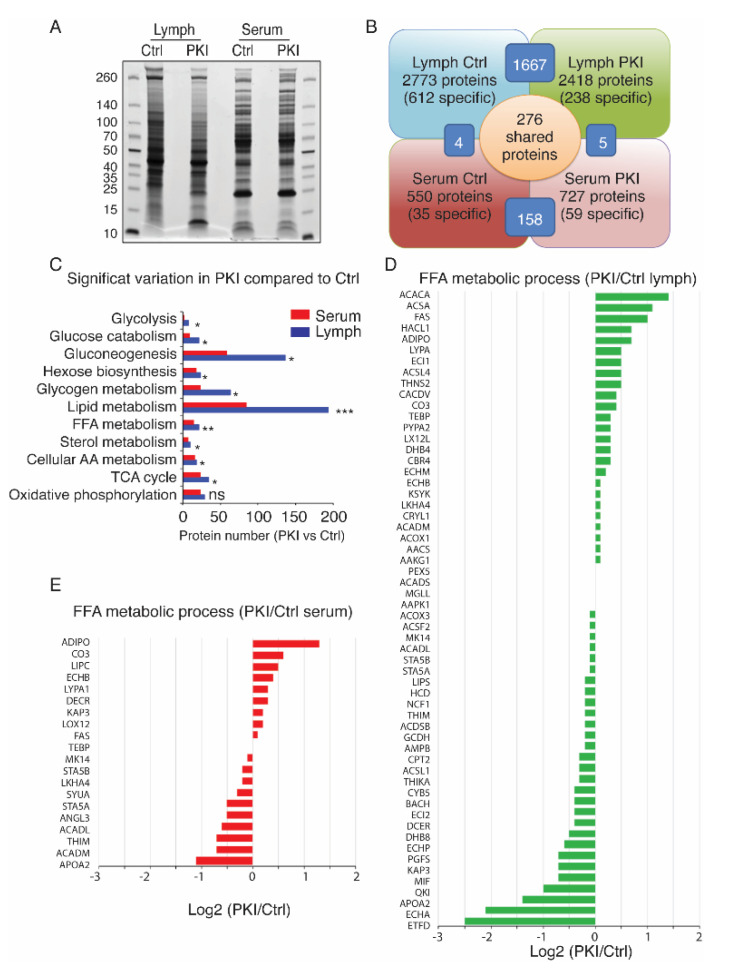
Proteomic analysis of the lymph and serum from tumor-bearing mice. (**A**) Lymph and serum peptides profile analysis in control and PKI mice after equalization. (**B**) Identified protein repartition in lymph and serum samples. (**C**) Mass spectrometry analysis of the lymph and serum from PKI mice exhibit significant changes in protein level in lymph compared to serum, in particular in proteins involved in the lipid metabolism process (mean + sem; *n* = nine animals; * *p* < 0.05, ** *p* < 0.01, *** *p* < 0.001). (**D**) Analysis of proteins involved in FFA metabolic process changes in lymph from PKI mice compared to control mice. (**E**) Analysis of proteins involved in FFA metabolic process changes in serum from PKI mice compared to control mice.

**Figure 5 cancers-13-02851-f005:**
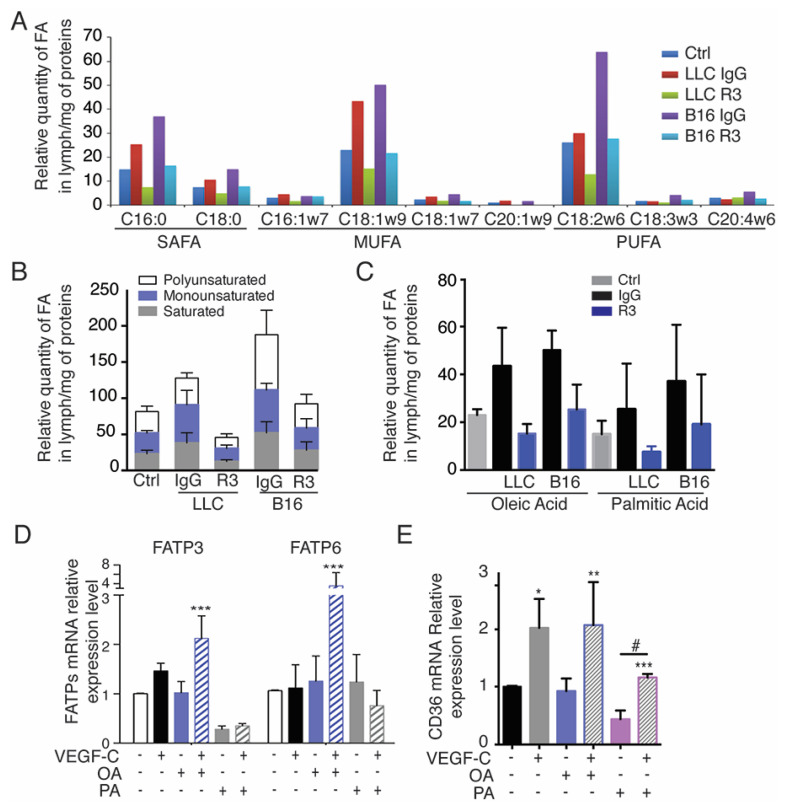
Lipidomic analysis of the lymph from tumor-bearing mice. (**A**,**B**) Lipidomic analysis of major saturated (SAFA), monounsaturated (MUFA) and polyunsaturated (PUFA) fatty acids in lymph from LLC and B16 tumor-bearing mice treated with IgG control isotype or anti-VEGFR-3 and compared to control mice (C16:0, palmitic acid; C18:0, stearic acid; C16:1w7, palmitoleic acid; C18:1w9, oleic acid; C18:1w7, vaccenic acid; C20:1w9, gondoic acid; C18:2w6, linoleic acid; C18:3w3, g-linoleic acid; C20:w6, arachidonic acid). (**C**) Relative quantification of Oleic Acid (OA, major monounsaturated FFA) and Palmitic Acid (PA, major polyunsaturated FFA) in lymph from LLC- and B16-tumor bearing mice treated with IgG control isotype or anti-VEGFR-3. (**D**) FATP-3 and 6 mRNA expression in OA or PA-stimulated LECs incubated with VEGFC (*** *p* < 0.001 compared to Ctrl-). (**E**) CD36 mRNA expression in OA or PA-stimulated LECs incubated with VEGFC (# *p* < 0.001, *** *p* < 0.001, ** *p* < 0.01, * *p* < 0.05 compared to Ctrl−).

**Figure 6 cancers-13-02851-f006:**
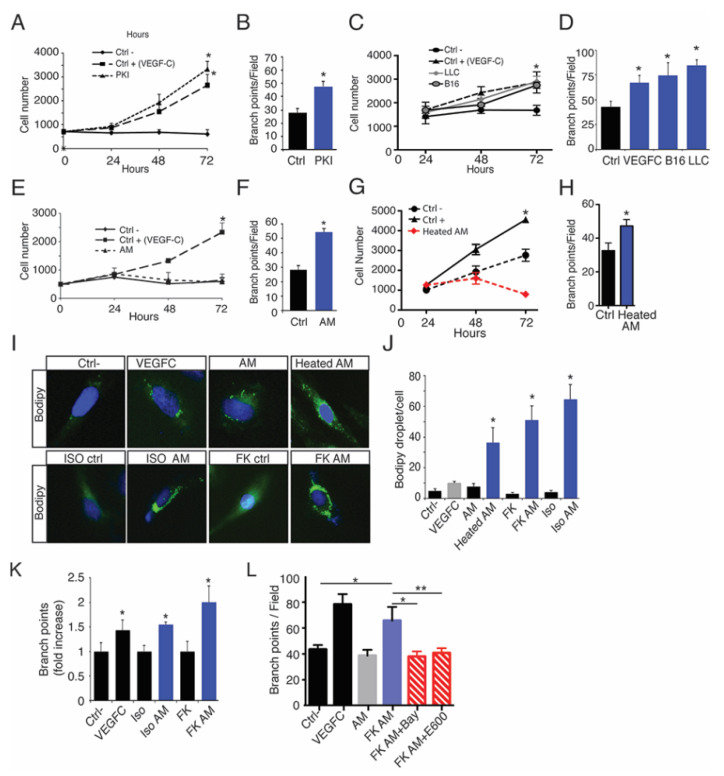
Exogenous FFA are uptaken by LEC to stimulate lymphatic sprouting. (**A**) Proliferation of PKI conditioned media-stimulated LECs (* *p* < 0.05). (**B**) Quantification of LEC branch point formation stimulated by PKI conditioned media (* *p* < 0.05). (**C**) Proliferation of LLC and B16 conditioned media-stimulated LECs (* *p* < 0.05). (**D**) Quantification of LEC branch point formation stimulated by LLC and B16 conditioned media (* *p* < 0.05). (**E**) Proliferation of LECs stimulated by Adipocyte Media (AM) (* *p* < 0.05). (**F**) Quantification of LEC branch point formation stimulated by AM conditioned media (* *p* < 0.05). (**G**) Quantification of heated AM-stimulated LEC proliferation (* *p* < 0.01). (**H**) Quantification of heated AM-stimulated lymphatic branch points formation (* *p* < 0.05). (**I**) Representative images of BODIPY lipid droplet uptake by LECs stimulated by VEGFC, AM, heated AM, and isoproterenol (ISO) or forskolin (FK) lipolysis-stimulated adipocyte conditioned media (scale bar, 50 μm). (**J**) Quantification of the number of BODIPY lipid droplet uptake by LECs stimulated by VEGFC, AM, heated AM, ISO- and FK-AM (* *p* < 0.05). (**K**) Isoproterenol (ISO) or forskolin (FK) lipolysis-stimulated adipocyte conditioned media stimulate LEC branch point formation (* *p* < 0.05). (**L**) Inhibition of LEC branch point formation in the presence of conditioned media from lipolytic adipocytes treated with hormone sensitive lipase (BAY), and non-selective lipase (E600) inhibitors (* *p* < 0.05; ** *p* < 0.01).

**Figure 7 cancers-13-02851-f007:**
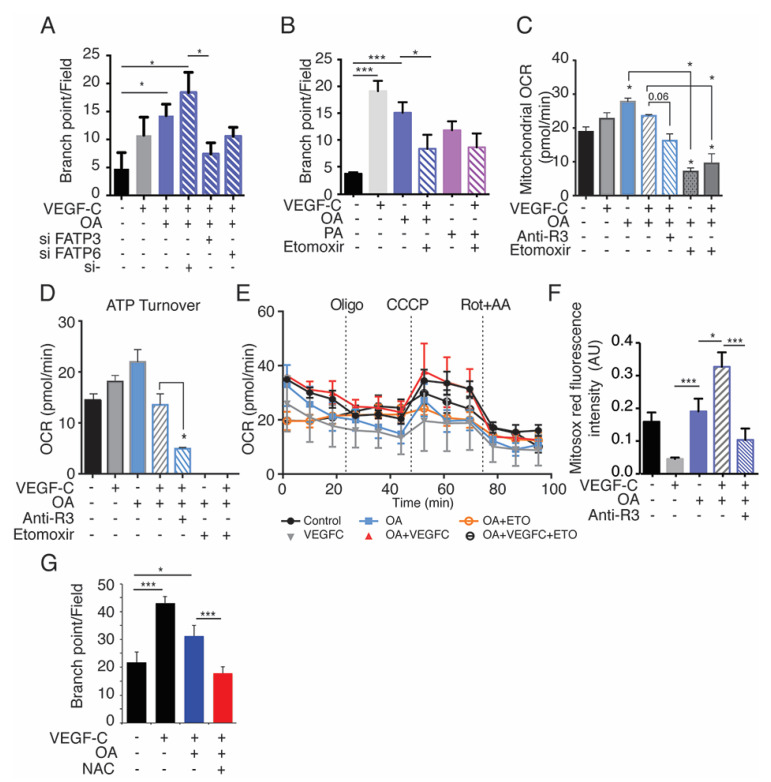
Oleic acid stimulates LEC FAO. (**A**) Quantification of OA+VEGFC stimulated lymphatic branch point formation in the presence of FATP3 and FATP6 siRNA (* *p* < 0.01). (**B**) Quantification of OA and PA-stimulated lymphatic branch point formation in the presence of etomoxir (*** *p* < 0.001, * *p* < 0.05). (**C**) OCR Mito stress test seahorse analysis for VEGFC+OA-stimulated LECs incubated with etomoxir or anti-VEGFR-3 (* *p* < 0.05). (**D**) OCR analysis showing ATP turnover in OA+VEGFC-stimulated LECs incubated with etomoxir (ETO) or anti-VEGFR-3 (R3) blocking antibody (* *p* < 0.05). (**E**) OCR Mito Stress test Seahorse profiles for OA+VEGFC-stimulated LECs incubated with etomoxir (ETO) or anti-VEGFR-3 (R3) blocking antibody. (**F**) Total fluorescent intensity of MitoSOX Red in LECs stimulated by VEGFC, OA, and anti-VEGFR3 (* *p* < 0.05; *** *p* < 0.001). (**G**) Inhibition of branch point formation by the N-acetyl cysteine (NAC) scavenger of ROS (* *p* < 0.05; *** *p* < 0.001).

## Data Availability

The data can be accessed from the corresponding author.

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
