# Peer review of "Coordinating Effect of VEGFC and Oleic Acid Participates to Tumor Lymphangiogenesis"

_cancers, 2021, doi:10.3390/cancers13122851_

Round 1

Reviewer 1 Report

In the study by Morfoisse et al the authors investigate whether adipose tissue derived fatty acids may participate in regulation of tumor lymphangiogenesis, which promotes tumor metastasis and reduces survival. Using human biopsies as well as multiple experimental models the authors convincingly demonstrate  1) lymphatic expansion in tumor-surrounding adipose tissue from premetastatic stage; 2) increased metastasis in tumors with peritumoral lymphangiogenesis; 3) increased leakiness into peritumoral adipose tissue of lymphatics in tumor-bearing mice; 4) increased protein and lipid content in lymph from tumor-bearing mice.

the authors go on to demonstrate selective increase in oleic acid and palmitic acid free fatty acids in the lymph of tumor-bearing mice with active lymphangiogenesis. Further, they demonstrate that VEGF-C upregulates fatty acid transporters FATP3 and FATP6 as well as CD36 on lymphatic endothelial cells (LEC), stimulating cellular lipid uptake. They also show that the non-heat sensitive components of adipocyte conditioned media stimulates lymphatic branching in vitro and increases LEC lipid uptake. Finally, they demonstrate that VEGFC or Oleic acid both increase LEC beta oxidation. Interestingly, only oleic acid, but not vegfc used alone, increased mitochondrial oxidative stress. They conclude that VEGFC and oleic acid may act in synergy to stimulate tumor lymphangiogenesis, providing a rationale for inhibition of lipolysis in addition to inhibition of VEGFC/VEGFR3 pathway to limit tumor metastasis.

comments:

in the experiment analysing lipid content in lymph from tumor-bearing animals treated or not with antiVEGFR3, was lipolytic hormones and cytokine (eg TNFa) plasma levels comparable in mice? Is it possible that antiVEGFR3, beyond just limiting lymphatic uptake locally, perhaps also acted systemically (by reducing metastasis burden) to reduce lipolysis to alter substrate availability in the tumor proximity for transport to lymph ?

please replace the word "unloaded" line 73, "upload" line 218 with "increased cellular FA uptake". 

in fig 1d, please replace PDAC with ADK for clarity

fig 2 legend, scalebar 50 micrometer not mm

in methods section or suppl methods please add information on histology and immunohistochemistry

fig 4a, please add kDa indication for protein sizes

Author Response

In the experiment analysing lipid content in lymph from tumor-bearing animals treated or not with antiVEGFR3, was lipolytic hormones and cytokine (eg TNFa) plasma levels comparable in mice?

We analyzed lipolytic hormones in lymph and demonstrated an increase in lipase level in tumor-bearing mice. This was reduced by the antiVEGFR3 treatment. Data are now provided in Supplementary figure 4A. However, no difference in TNFa expression was observed at preneoplastic stages.

As observed in Figure 4 A and B there is little change in the protein profile between Ctrl and PKI mice in the serum compared to the lymph (1167 compared to 158). This is further strengthened by the quantification of ApoA protein family in serum and lymph: while we observed an increase in the lymph of tumor-bearing mice compared to control, no change has been observed in serum. Data are now presented in Supplementary figure 5A and B.

While it is still possible to have differences in plasmatic levels of lipolytic hormones between control and tumor-bearing mice, we feel that this is out of the scope of this study as we choose to focus our work on the crosstalk between tumors cells, lymphatic vessels and adipose tissue.

Is it possible that antiVEGFR3, beyond just limiting lymphatic uptake locally, perhaps also acted systemically (by reducing metastasis burden) to reduce lipolysis to alter substrate availability in the tumor proximity for transport to lymph?

We thank the reviewers for bringing up this interesting point. Indeed, we performed Echo-MRI quantification of the fat mass in LLC and B16 tumor-bearing mice and observed an increase of the fat mass with the antiVEGFR3 treatment. In addition, we demonstrated that the antiVEGFR3 treatment inhibits tumor-distant (axillary) AT loss in LLC-bearing mice. These results advocate for a systemic inhibition of lipolysis induced by the antiVEGFR3 treatment. Data are now presented in Supplementary figure 4B and C.

please replace the word "unloaded" line 73,

“Unloaded” was replaced by “loaded”

"upload" line 218 with "increased cellular FA uptake". 

Modification has been performed.

in fig 1d, please replace PDAC with ADK for clarity

Modification has been performed.

fig 2 legend, scalebar 50 micrometer not mm

Modification has been performed.

in methods section or suppl methods please add information on histology and immunohistochemistry.

As requested by the reviewer, we added the following experimental details in methods section :

Immunohistochemistry

Tumors, lymph nodes and adipose tissues were embedded into OCT compound and 5 µm tissue sections were immunostained with specific antibodies. The mean number of lymphatic vessels was quantified in 5–10 microscopic fields per cryosection using automated pixel density determination. The mean number of mice with metastases in draining lymph nodes was determined by immunostaining cryosections with 10 μg/mL of anti-pancytokeratin (Santa Cruz Biotechnologies, Dallas, TX, USA). Immunostaining was quantified in 5–10 microscopic fields per cryosection using automated pixel density determination as the mean number of pixels +/- SEM for each group.

fig 4a, please add kDa indication for protein sizes

Molecular weight (kDa) indication for protein sizes have been added to figure 4A

Reviewer 2 Report

The author proposed adipocyte derived FFA promotes lymphangiogenesis. The story is very interesting, but the data and evidence is weak and not sufficient to support the conclusion. The assay to collect lymph which is the very key part of the whole paper is questionable.

Here are some comments:

1.The survival rate is 3 fold increase when blocking VEGFR3; however, the lymph node metastases is changed slightly (4/8 or 6/8 from 8/8 and 6/6). Does it make sense? Blocking VEGFR3 to improve survival may be contributed by other mechanisms in addition to lymphatic metastasis. If yes, the metastasis facilitated by lymphatics is really key?

2.The way to collect lymph from lymph node used here is tricky.
There is no way to control the size of lymph nodes and it also doe not know whether lymph nodes/fluid in lymph nodes from controls or tumor conditions are the same or not. Therefore, the volume of lymph from lymph node is not standardized. I have a concern about the key methodology here.

3.In fig3, it shows that there is a leakage. The lymph contains lipids, metabolites, HDL cholesterol…etc in addition to FFA. It is hard to conclude the lymphangiogenesis is only contributed by FFA.
There might exist a correlation, but the direct evidence is lacking. Plus, the way to collect lymph to quantify FFA is not solid (comment above).

4.In fig4, there is a proteomic data to correlate the FFA pathway, but what does that mean? I have no idea about the info provided in fig4D and E. What do those proteins in figs demonstrate? Besides, lymph contains a lot of things draining from interstitium. There must be proteins related to lipids, but how the data provided here helps to demonstrate the idea, FFA for example?

5.In fig 5, it is an in vitro culture exp and can not completely reflect the in vivo situation. Again, the data can not explain if FFA is even increased in vivo, the question raised from the comment above. The author might consider to use CD36 KO mice in lymphatics to demonstrate the importance of FFA to lymphatics and lymphangiogenesis in vivo.

6.In fig6, the author showed that FFA stimulates lymphatic spouting in vitro. Again, what does that mean to explain the in vivo tumor situation? Did the author also notice the sprouting of lymphatics under tumor conditions? How does FFA control spouting? If the author did not notice the sprouting in vivo, how does this in vitro data help the story?

7.In fig7, this is an in vitro exp and I don’t understand the results from anti-R3. It is understandable in vivo when treating anti-R3, lymphangiogenesis is reduced such that FFA is reduced and the tumor burden is less. When doing in vitro culture exp, as long as there is OA, OCR should be increased no matter if there is an anti-R3 or not. The author noticed OCR is down under anti-R3 and the explanation as I see it is that the cell number is less, which is not the same mechanism the author would like to address in vivo. I kind of get a bit lost based on my understanding of the draft.

Author Response

1.The survival rate is 3 fold increase when blocking VEGFR3; however, the lymph node metastases is changed slightly (4/8 or 6/8 from 8/8 and 6/6). Does it make sense? Blocking VEGFR3 to improve survival may be contributed by other mechanisms in addition to lymphatic metastasis. If yes, the metastasis facilitated by lymphatics is really key?

We thank reviewer 2 for his careful evaluation. When blocking VEGFR3, we still observed some metastases into the lymph nodes. However, the metastases size is significantly reduced compared to isotype treated mice suggesting that there is a strong delay in the occurrence of metastatic process. Also, VEGFC/VEGFR3 is not the unique pathway involved in tumor lymphangiogenesis that can be stimulated by other growth factors such as FGF2 or HGF (Tammela T., Cell 2010) that can explain this residual phenotype.

This result support our message showing that tumor lymphangiogenesis not only provides routes for metastasis, but is also involved in other process such as lipid transport including lipase that control distant adipose depots weight. The delay in fat loss might be associated to the survival improvement as described by Das SK (Science 2011).

These data are now added in supplementary figure 4.

2.The way to collect lymph from lymph node used here is tricky.
There is no way to control the size of lymph nodes and it also doe not know whether lymph nodes/fluid in lymph nodes from controls or tumor conditions are the same or not. Therefore, the volume of lymph from lymph node is not standardized. I have a concern about the key methodology here.

The major option to collect lymph in mice is to canulate the cisterna chyli (M. Nagahashi et al. 2016). However, the only possibility to collect the lymph is in postprandial phase when the lymph is full of alimentary lipids. In order to study lipid components of the lymph, we had to extract lymph from another loci. We collected lymph form tumor draining lymph nodes that appeared to be the most representative of the tumor microenvironmental fluids.

We weighted the sentinel lymph nodes, but no significant difference was observed. We collected 2-3uL of lymph per lymph nodes in 9 mice per group. All the data were normalized by the total protein content for each mouse (proteomic section in the methods). The protein changes observed in PKI compared to control mice were performed using the same amount of protein in each group.

3.In fig3, it shows that there is a leakage. The lymph contains lipids, metabolites, HDL cholesterol…etc in addition to FFA. It is hard to conclude the lymphangiogenesis is only contributed by FFA.

We agree with reviewer 2, the lymph transport other metabolites than FFAs, in particular cholesterol transporters such as apolipoproteins. We observed an increase of apolipoproteins in lymph from tumor bearing mice compared to control, but not in serum, showing changes in metabolic processes in this vascular compartment at preneoplasic stages. Data are now available in supplementary figure 5. However, despite a clearly demonstrated role of the lymphatic system in the reverse cholesterol transport (Lim HY, Cell Metab 2013), the effect of cholesterol in lymphangiogenesis remains undetermined.

In addition, Escobedo N. and colleagues (JCI Insight 2016) described the crucial role of lymph FFA in promoting adipocyte differentiation and their role in the maturation of adipocytes.

Altogether, figure 3 supports the hypothesis that lymphatic system transport of peripheral lipids is increased in tumor development and can be reversed by the anti-VEGFR-3 antibody. The direct effect of FFA on lymphatic endothelial cells in demonstrated lately in figures 6 and 7 of the manuscript.

There might exist a correlation, but the direct evidence is lacking. Plus, the way to collect lymph to quantify FFA is not solid (comment above).

We could not agree with this point as the lymph analysis has been normalized by the total protein contents. Significant variations were observed by the data analyst from the protemic platform of genotoul that is independent from our laboratory.

4.In fig4, there is a proteomic data to correlate the FFA pathway, but what does that mean?

The lipid metabolic process led to the activation of lipolytic enzymes that generate the release of FFAs to provide energy for the body. In figure 4, we observed differences in the lymph at neoplastic stages, which means before the tumor develops. These data suggest that lipid metabolic changes occur before the adenocarcinoma stages to probably prepare the carcinogenic process. Importantly, this mechanism is only observable in the lymphatic system  (not in serum) that collects tissue fluids and is thus representative of the microenvironment status.

I have no idea about the info provided in fig4D and E. What do those proteins in figs demonstrate?

Fig 4D and E show the modifications of lymph proteins involved in FFA metabolism in PKI mice compared to control littermates. It shows that modification in protein content observed in serum (19 proteins) is a lot less important compared to the lymph (54 proteins). This highlights the importance to do no restrict the fluid analysis to the serum because stronger changes are observed in lymph.

Besides, lymph contains a lot of things draining from interstitium. There must be proteins related to lipids, but how the data provided here helps to demonstrate the idea, FFA for example?

We agree with reviewer, the Figure 4 does not provide any proof showing that FFA stimulate lymphatic endothelium. However, we found a significant increase of lipase that generate the release of FFA. Data are now provided in supplementary figure 4A. All changes observed in lipid metabolic process led us to perform lipidomic analysis as shown in figure 5. This figure clearly established modifications in FFAs content in the lymph at preneoplastic stages.

5.In fig 5, it is an in vitro culture exp and can not completely reflect the in vivo situation. Again, the data can not explain if FFA is even increased in vivo, the question raised from the comment above. The author might consider to use CD36 KO mice in lymphatics to demonstrate the importance of FFA to lymphatics and lymphangiogenesis in vivo.

Reviewer 2 might have misunderstood figure 5 as it is an “in vivo” lipidomic analysis of the lymph and not “in vitro”. Figure 5 is perfectly representative of the in vivo study and shows that all the FFA levels are increased in the  tumor-bearing mice lymph (SAFA, MUFA, PUFA), in particular the oleic acid. More importantly, this process is reversed by the anti-VEGFR-3 blocking antibody.

Also, generating a CD36 conditional lymphatic KO mice would have been an interesting point. These mice would bring additional informations for the manuscript, however as shown in figure 5, CD36 is not the only one receptor responsible for FFA transport into the lymphatic endothelial cells.

Therefore, the conditional knock out of CD36 may not be sufficient to inhibit the loading of oleic acid in lymphatics. Also, these conditional KO mice are not available and it would take more than 1 year to generate it.

6.In fig6, the author showed that FFA stimulates lymphatic spouting in vitro. Again, what does that mean to explain the in vivo tumor situation?

Figure 6 aims to investigate what would be the consequence of an increase of FFA (observed before) on the lymphatic endothelium.

In figure 1 and 2, we showed that lymphangiogenesis develops at the interface between lesions and adipose tissue before the adenocarcinoma (tumoral) stage suggesting that something else than tumor-synthesized growth factors is involved in lymphangiogenesis. In fig 6 we showed that FFA can participate to the stimulation of lymphangiogenesis independently of growth factors.

Did the author also notice the sprouting of lymphatics under tumor conditions?

We observe tumor-induced sprouting, which is the first step of lymphangiogenesis, of lymphatic vessels in vivo and in vitro. In vivo data are shown in figure 1E and F,  in vitro data are shown in fig 6B (PKI) and 6D (LLC, B16).

How does FFA control spouting? If the author did not notice the sprouting in vivo, how does this in vitro data help the story?

Lymphatic sprouting is dependent on intra-cellular fatty acids (FA) beta-oxidation (Wong BW, Nature 2016). Here, we demonstrated that extra-cellular Free Fatty Acids (FFA) could also participate to that metabolic process after being uptaken by LECs.

We observe the sprouting in vivo before the adenocarcinoma (tumoral) stage suggesting that FFA could be the alternative stimulator of lymphangiogenesis. In fig 6 we showed that FFA alone are able to stimulate LEC sprouting after being loaded.

7.In fig7, this is an in vitro exp and I don’t understand the results from anti-R3. It is understandable in vivo when treating anti-R3, lymphangiogenesis is reduced such that FFA is reduced and the tumorz burden is less. When doing in vitro culture exp, as long as there is OA, OCR should be increased no matter if there is an anti-R3 or not. The author noticed OCR is down under anti-R3 and the explanation as I see it is that the cell number is less, which is not the same mechanism the author would like to address in vivo. I kind of get a bit lost based on my understanding of the draft.

In Fig 5D we showed that OA alone was not able to stimulate its transporter FATP3, FATP6 and CD36 expression. OA effect has to be coordinated with VEGFC to promote lymphatic sprouting.  Using the anti-R3, the expression of OA transporters is inhibited and thus the OCR does not increase. Also, we have to pay attention that these results have been obtained on primary cultures of human dermal lymphatic endothelial cells, which do not have a strong basal metabolism as observed in cultured immortalized or tumoral cell lines.

Reviewer 3 Report

In this manuscript, authors examined the tumor lymphangiogenesis in different mouse cancer models and human pancreatic ductal adenocardinoma (PDAC) using podoplanin, LYVE1 as well as lectin as markers of lymphatic vessels. Authors concluded that tumor lymphangiogenesis developed in tumoral lesions and in their surrounding adipose tissue (AT), and adipocyte-derived free fatty acids (FFA) play a key role in the pro-motion of lymphangiogenesis.

Several issues need to be addressed.

Authors used a blocking antibody against VEGFR-3 and VEGFR-2 to study the molecular mechanisms of peritumoral lymphangiogenesis. It would be great if authors could show some data to indicate that VEGFRs were involved in lymphangiogenesis in the samples, e.g. the expression of VEGFRs upregulated in the tumor samples compared to those obtained from normal control.   

Various lymphatic vessel markers were used to quantify the lymphangiogenesis in the different samples obtained from human and mice. In addition, lectins are carbohydrate-binding proteins. It seems not very good to be used as a vessel marker in Figure 3A and 3B. 

The data shown in Figure 1D seems not match well with Figure 1C, since the sample of PDAC was obtained from human.

Figure 4C shows the significant variation in protein level in lymph compared to serum from PKI mice. It would be better if authors can show the similar analysis of proteins in lymph compared to serum in normal control mice.

Author Response

Authors used a blocking antibody against VEGFR-3 and VEGFR-2 to study the molecular mechanisms of peritumoral lymphangiogenesis. It would be great if authors could show some data to indicate that VEGFRs were involved in lymphangiogenesis in the samples, e.g. the expression of VEGFRs upregulated in the tumor samples compared to those obtained from normal control.   

We performed RT-qPCR analysis of VEGFR3 expression in tumors. We observed an increase of VEGFR-3 expression during tumor progression.

Various lymphatic vessel markers were used to quantify the lymphangiogenesis in the different samples obtained from human and mice. In addition, lectins are carbohydrate-binding proteins. It seems not very good to be used as a vessel marker in Figure 3A and 3B. 

While we agree with the reviewer that lectins are not the gold standard to detect vessels in immunostainings, we would like to emphasize that in our study the FITC-lectin was injected in the footpad of the mice leading to an intralymphatic diffusion. This injection technique has been demonstrated by our group and others (Morfoisse et al, ATVB 2018) to be reliable to perform intra-lymphatic injection of fluorescent compound. Interestingly we quantified fluorescent intensity in both LLC and B16 tumors after injection of FITC lectin and observed a decrease of fluorescence intensity induced by the antiVEGFR3 treatment. This strongly contributes to support that FITC-lectin particles injected in the footpad are directly taken up by the lymphatic circulation. Data are now provided in Supplementary figure 3A and B.

The data shown in Figure 1D seems not match well with Figure 1C, since the sample of PDAC was obtained from human.

Figure 1C and D correspond to histological analysis of murine tissues from PKI mice. This part of the figure aims to highlight that the lymphatic vessel infiltration starts at preneoplastic stages, in particular PanIN3 stage (before adenocarcinoma).

Figure 4C shows the significant variation in protein level in lymph compared to serum from PKI mice. It would be better if authors can show the similar analysis of proteins in lymph compared to serum in normal control mice.

Figure 4C shows the significant variation in protein level in lymph and in serum from PKI mice compared to control mice. This point was unclear and the figure legend has been modified for a better understanding.

Round 2

Reviewer 2 Report

  1. In response to #1, what is not clear to me is that blocking VEGFR3 pathway reduces lymphatic vessels and increases survival, but slightly changes metastases, suggesting that the death is not mainly contributed by metastases through lymphatic vessels, but others. The author believe that the results support their conclusion in that the improved survival is contributed by the lipid transport. If yes, then the following data in the draft is required to be solid to support the statement. There is a lot of data and exps to support the statement, but they are all correlative, which is very good, but lack of direct evidence.
  2. In response to #4, it is now interesting to see the supplemental data fig4 that lipase is increased, supporting the significance of FFA in lymph. Is the lipase activity up too? Traditionally, lipase is anchored in the endothelium where FFA is generated and updated by cell right away. As suggested, lipase level, not activity, is up in lymph, does that also mean lipase attaches more to the lymphatic endothelium? The author should treat mice with lipase inhibitor to check if FFA in lymph is reduced w/wo anti-R3 and correlates with the mouse survival and metastasis. This exp will strongly support the idea that FFA in lymph, lymphangiogenesis and tumor progression.
  3. In response to #5, sorry to be unclear, fig 5D is an in vitro exp correct? It is a pretty interesting observation, but may not represent the in vivo situation. CD36 condition KO is available and published (please check the paper). It is true tho that even the mouse exists, it will take a while. But then the study here is still correlative.
  4. Overall the draft is very interesting and I am positive to see the work to be published. But the data is not strong enough to support the conclusion. The title of the paper is too strong without providing further causal-effect evidence. The author should consider to play down a bit of the title.

Author Response

  1. In response to #1, what is not clear to me is that blocking VEGFR3 pathway reduces lymphatic vessels and increases survival, but slightly changes metastases, suggesting that the death is not mainly contributed by metastases through lymphatic vessels, but others. The author believe that the results support their conclusion in that the improved survival is contributed by the lipid transport. If yes, then the following data in the draft is required to be solid to support the statement. There is a lot of data and exps to support the statement, but they are all correlative, which is very good, but lack of direct evidence.

Patient survival is highly correlated to the metastatic dissemination. However, we agree with reviewer 2 as in some cancers such as pancreatic ductal adenocarcinoma, melanoma and lung carcinoma, another parameter influences the outcome of the patient: the weight loss called cachexia. Once the patient starts to lose weight, he becomes resistant to chemotherapies and the survival prognosis is poor.

Weight loss is a characteristic of gastrointestinal cancers, develops in 80% of patients with a high prevalence in Pancreatic Ductal AdenoCarcinoma (Tisdale, 2010; Zechner et al., 2012). Furthermore, cancer cachexia is a multifactorial syndrome associated with anorexia, skeletal muscle, and adipose tissue lipolysis that can lead to death. The adipose tissue reduction is correlated with lipolysis and involves two major enzymes, the Adipose TriGlyceryde Lipase (ATGL) and the Hormone Sensitive Lipase (HSL)(Das et al., 2011). In cancer, an exacerbated adipocyte lipolysis favors weight loss and emaciation (Arner and Langin, 2014). Cancer-associated lipolysis generates high levels of lipids that can be detected in the lymph. Deletion of ATGL and HSL reduces fatty acid mobilization, retains adipose tissue mass, and prevents weight loss (Das and Hoefler, 2013). Here, we found a decrease in gat mass and in adipose depots mass during tumor development that is reversed by the anti-VEGFR3 antibody (Supplementary Figure 4B and C).

The main physiological role of lymphatic vessels is to drain lipids, and most of the reported pathologies involving lymphatic vessels and adipose tissue result in fat accumulation. Defective lymphatic endothelial cell junctions abrogate weight gain under high-fat diet and fat accumulation was observed in mice that exhibit genetic or lymphatic dysfunction (Engberg et al., 2015; Greene et al., 2012). Besides these examples, it remains unknown if and how lymphatic vessels control the drainage of adipose depots in cancer.

This study may represent the first step to identify a role of the lymphatic system in the control of adipose depots weights by the transport of FFA from tumor-proximal to -distant tissues. However, the direct link between lymphatic growth and cachexia is not identified in this manuscript. We aimed to demonstrate that an increase of circulating oleic acid in cancer participates to tumor lymphangiogenesis, which contributes to a poor prognosis through both increased lymphatic dissemination and fat loss induction.

  1. In response to #4, it is now interesting to see the supplemental data fig4 that lipase is increased, supporting the significance of FFA in lymph. Is the lipase activity up too? Traditionally, lipase is anchored in the endothelium where FFA is generated and updated by cell right away. As suggested, lipase level, not activity, is up in lymph, does that also mean lipase attaches more to the lymphatic endothelium?

We observed an increase of lipase levels in lymph, not at the lymphatic endothelium surface. This is linked to an increase of Free Fatty Acid levels in lymph, in particular Oleic Acid, which support the idea that lipase activity is increased (Figure 5). Moreover, the inhibition of lipase activity completely abolished the lymphangiogenic effect induced by adipocytes conditioned media on lymphatic endothelial cells. While performed in vitro, this result again points to an increased lipase activity during tumor progression in vivo.

  1. The author should treat mice with lipase inhibitor to check if FFA in lymph is reduced w/wo anti-R3 and correlates with the mouse survival and metastasis. This exp will strongly support the idea that FFA in lymph, lymphangiogenesis and tumor progression.

We agree with reviewer 2, treating mice with lipase inhibitor would be interesting to evaluate the role of FFA on lymphangiogenesis. However, lipase inhibitors generate an accumulation of adipose tissue leading to weight gain. In that context, we may observe a phenotype attributed to the overweight that is demonstrated as deleterious in tumoral context.

More importantly, the invalidation of lipases in tumoral context has been previously performed by Das and colleagues (Science 2011). In this study, the authors used the B16 and LLC models that we are currently using in our manuscript. They observe that inhibition of lipase in mice reduces cancer-associated cachexia and significantly improved survival.

  1. In response to #5, sorry to be unclear, fig 5D is an in vitro exp correct? It is a pretty interesting observation, but may not represent the in vivo situation. CD36 condition KO is available and published (please check the paper). It is true tho that even the mouse exists, it will take a while. But then the study here is still correlative.

The expression of FA transporters has been performed on primary cultures of human lymphatic endothelial cells (passage 2 to 4), which are the best model to study molecular regulations observed in the lymphatic system. As we previously explained, CD36 in not the major FFA transporter on LEC surface. CD36 is increased by VEGFC, but no more stimulation of expression is observed with OA+VEGFC whereas FATP3 and FATP6 are significantly overexpressed by OA+VEGFC.

Also, CD36 binds to collagen, thrombospondin, anionic phospholipids and oxidized LDL. Therefore, invalidating CD36 in the lymphatic endothelium would bring a lot’s of information than a process associated only to the lack of oleic acid transport.

Here, we 1/ identified FATP3 and FATP6 (2 and 6 in blood endothelium) as the main FFA tyransporters and 2/ invalidated them in lymphatic endothelial cells to evaluate the direct effect. We confirmed a selective effect of oleic acid on the lymphatic endothelium that is not only correlative.

  1. Overall the draft is very interesting and I am positive to see the work to be published. But the data is not strong enough to support the conclusion. The title of the paper is too strong without providing further causal-effect evidence. The author should consider to play down a bit of the title.

We thank reviewer 2 for the general positive comment. We propose to change the current title “Coordinating effect of VEGFC and Oleic Acid drives tumor lymphangiogenesis” for

“Coordinating effect of VEGFC and Oleic Acid participates to tumor lymphangiogenesis”

Reviewer 3 Report

Authors used a blocking antibody against VEGFR-3 and VEGFR-2 to study the molecular mechanisms of peritumoral lymphangiogenesis. It would be great if authors could show some data to indicate that VEGFRs were involved in lymphangiogenesis in the samples, e.g. the expression of VEGFRs upregulated in the tumor samples compared to those obtained from normal control.   

We performed RT-qPCR analysis of VEGFR3 expression in tumors. We observed an increase of VEGFR-3 expression during tumor progression.  

I appreciate authors who have seriously considered my comments and performed some experiments.  However, the figure authors listed in response seems not show any significant differences in the expression of VEGRR3 between two groups.

Figure 4C shows the significant variation in protein level in lymph compared to serum from PKI mice. It would be better if authors can show the similar analysis of proteins in lymph compared to serum in normal control mice.

Figure 4C shows the significant variation in protein level in lymph and in serum from PKI mice compared to control mice. This point was unclear and the figure legend has been modified for a better understanding.

This reviewer still can’t understand the detail information in Figure 4C. If it represents the protein numbers in Serum and Lymph obtained from PKI and control, each (e.g. Lipid metabolism) should have 4 protein numbers, i.e. 1) Control serum, 2) Control Lymph, 3) PKI serum, and PKI Lymph, respectively.  

Author Response

  1. Authors used a blocking antibody against VEGFR-3 and VEGFR-2 to study the molecular mechanisms of peritumoral lymphangiogenesis. It would be great if authors could show some data to indicate that VEGFRs were involved in lymphangiogenesis in the samples, e.g. the expression of VEGFRs upregulated in the tumor samples compared to those obtained from normal control.   

We performed RT-qPCR analysis of VEGFR3 expression in tumors. We observed an increase of VEGFR-3 expression during tumor progression.  

I appreciate authors who have seriously considered my comments and performed some experiments.  However, the figure authors listed in response seems not show any significant differences in the expression of VEGRR3 between two groups.

We agree with reviewer 3, the difference in VEGFR3 expression is not significantly increased in tumors compared to control. However, this measure is not fully representative of VEGFR3 expression on lymphatic endothelial cells as it is also expressed by blood angiogenic sprouts (Tamela, Nat Cell Biol 2008) and macrophages (Alishekevitz Cell Rep. 2016). However, we showed that angiogenesis is not the major route to convey lipids as no difference was observed in serum compared to lymph (Figure 4 and supplementary figure 5). Concerning the immune cells, despite an increase of immune infiltration in tumors (as observed in almost all the solid tumors), we could not find any effect of the anti-VEGFR3 blocking antibody (see below).

In contrast, when measuring tumor lymphangiogenesis using two different markers (Lyve-1 and Podoplanin), we observed and increase in lymphatic vessel density. Altogether, these data led us to converge to a lymphatic endothelial selective effect of the VEGFR3 blocking antibody.

For VEGFR2 expression, it is the predominant mediator of VEGF-induced angiogenesis in cancer and its blockade has been extensively investigated. Anti-VEGFR2 are currently used for the treatment of solid tumors (kidney, breast, lung,…) starting with the sunitinib (6560 publications). Therefore, we made the editorial choice not to burden the manuscript with data concerning the development of angiogenesis in tumors as our results associated with FFA were related to the lymphatic, but not blood vessels.

  1. Figure 4C shows the significant variation in protein level in lymph compared to serum from PKI mice. It would be better if authors can show the similar analysis of proteins in lymph compared to serum in normal control mice.

Figure 4C shows the significant variation in protein level in lymph and in serum from PKI mice compared to control mice. This point was unclear and the figure legend has been modified for a better understanding.

This reviewer still can’t understand the detail information in Figure 4C. If it represents the protein numbers in Serum and Lymph obtained from PKI and control, each (e.g. Lipid metabolism) should have 4 protein numbers, i.e. 1) Control serum, 2) Control Lymph, 3) PKI serum, and PKI Lymph, respectively.  

We apologize for this misunderstanding. Fig 4C graph shows the number of significant variants in lymph and in serum from tumor bearing mice compared to normal mice.

Therefore, we showed 2 protein numbers as it represent only the number of variations in tumoral condition compared to normal.

We showed that in lymph, there is 29 proteins involved in glycolysis identified with significant change in tumoral condition, 34 proteins involved in glucose catabolic process identified with significant change in tumoral condition, etc…

We sincerely hope this explanation will help for a better understanding of reviewer 3.

Round 3

Reviewer 2 Report

none